# A New Calculation Method of Force and Displacement of Retaining Wall and Slope

**Yingfa Lu** [1,*], **Wenqing Sun** [1], **Hao Yang** [1], **Junjie Jiang** [1] and **Lier Lu** [2]

1   School of Civil Architecture and Environment, Hubei University of Technology, Wuhan 430068, China; wsswq1996@foxmail.com (W.S.); qkszl777@foxmail.com (H.Y.); j18727228579@163.com (J.J.)
2   School of Engineering, China University of Geosciences (Wuhan), Wuhan 430074, China; m13349992057@126.com
*   Correspondence: lyf77@126.com; Tel.: +86-138-071-276-73

**Abstract:** On the basis of the force and safety factor analysis of traditional retaining walls, a new analytical method of force and displacement of the slope is suggested, the numerical theoretical solution of the stress distribution of the sliding body can be obtained under the conditions, which the stresses distribution are satisfied with the differential equilibrium equations, the boundary conditions, the compatibility equation and the macroscopic equilibrium equations. The interface stresses between the sliding body and the retaining wall is continuous, and the theoretical solution of retaining wall stress distributions can be obtained, while the stress differential equilibrium equation, the compatibility equation, macroscopic force, and moment balance equations are satisfied. The strain and displacement solutions can be obtained by using Duncan Chang and Hooke constitutive equations for the slope and the retaining wall, respectively. The transfer station of landfill in the Guandukou Town of Badong County is taken as an example, the results of the sliding body and retaining wall analysis show: The stress and strain solutions of a slope and a retaining wall can be obtained by the proposed method. The anti-slip force of the retaining wall calculated by the method in this paper contains the positive pressure and shear force along the contact surface and varies with the deformation of the slope, in addition, the numerical theoretical solution of the retaining wall shows that the retaining wall shape and material can be optimized according to the calculation results. It is feasible for the proposed analysis method of slope with retaining wall design to be run many years.

**Keywords:** numerical theoretical solution; equilibrium equation; stress and strain distribution; slope; retaining wall



## 1. Introduction

A retaining wall is a structure used to support roadbed fill or hillside soil to prevent the deformation and destabilization of the fill or soil. Retaining walls can stabilize embankment and graben slopes, reduce the height of excavated slopes, reduce the amount of earth and stone excavation and floor space, protect the foot of roadbed slopes, prevent water from washing the roadbed, prevent the slope cover from sliding, and prevent landslides, and they are often used for the remediation of landslides and other disasters.

To reduce the risk of hazards arising from damaged slopes and retaining walls, effective protection techniques have been proposed on the basis of stability analysis of slopes and retaining walls, and a series of studies have been conducted by domestic and foreign scholars mainly in terms of theoretical research, numerical analysis, and model tests. Chen [1] introduced the design points of various retaining walls and the force characteristics and summarized the advantages and disadvantages. Zhao [2] discussed the design process of weighted retaining walls under the influence of various factors, such as groundwater and fill material behind the wall. Li [3], from drain simulation theory, established three-dimensional steady and unsteady seepage models of the fill behind retaining

walls considering the action of drains. Xu [4] experimentally simulated the soil pressure distribution law of retaining walls when sandy soil was used as fill. Dawson EM, Roth WH, and DrescherA [5] established a computational model that integrates retaining wall and slope information and analyzed the stability of the slope of a highway section with limit equilibrium theory and finite element software.

The active earth pressure calculation method for retaining walls is a traditional topic in the field of geomechanics. To date, there are problems in the accurate calculation of the active earth pressure, which are mainly reflected in the magnitude of the active earth pressure, the location of the action point, the deformation and damage mode of the retaining wall, the need to make assumptions about the sliding and cracking surface, and the established calculation model resulting in limited applicability. Coulomb earth pressure theory assumes that a sliding surface is flat, the fill behind the wall is cohesionless soil, and the earth pressure is triangularly distributed to obtain the active earth pressure calculation formula. Rankin's earth pressure theory takes semi-infinite space soil as the object of study and assumes that the wall is rigid, the back of the wall is vertical and smooth, the surface of the fill behind the wall is horizontal, and the earth pressure distribution is triangular, and the corresponding theoretical solution is obtained from the ultimate equilibrium state. For cohesive soil, the pressure can be calculated directly by Rankin's theory, but the results are too conservative. Terzaghi [6] considers that the active earth pressure on a rigid retaining wall is related to the form of wall motion, and there are obvious differences in its active earth pressure under three different modes of motion: the wall is moving horizontally and rotating around the top and bottom of the wall. Mao [7] explained the deficiencies of the Coulomb earth pressure calculation theory from a mechanical point of view and clarified that the wall surface and slip crack surface could not reach the ultimate equilibrium state at the same time. Kezdi [8] studied the motion mode of a retaining wall rotating around the bottom of the wall. Handy RL [9] applied the basic principle of the soil arch effect, assumed the slip crack surface to be a Rankin slip crack surface, and combined it with the differential unit method to study the stress distribution of the soil behind the retaining wall. Wang and Sun [10] improved the active earth pressure assumption and proposed a new calculation method. Chen [11] analyzed the stress state of a retaining wall by taking three basic deformation modes of the wall as the research object. He [12] studied the calculation method of layered soil.

At present, the stability of retaining walls is mainly evaluated to maintain safety, and although it has been applied in engineering practice for a long time, some retaining walls can still be damaged when the shear and overturning strengths are satisfied. Zeng and Zhou [13] analyzed the types of overturning failures of retaining walls and concluded that the overturning resistance of a retaining wall is related to the ultimate bearing capacity of the foundation. Gan et al. [14] analyzed the problems in the expressions of the overturning stability equation in the current code. Huang et al. [15] studied the overturning stability of retaining walls under earthquake action by the proposed dynamic method. Huang [16,17] noted that the traditional theory is too simple.

With the development of numerical analysis methods, different calculation methods have been proposed [18–23], and recently, the partial strength reduction method for progressive damage processes has been widely applied.

## 2. Research and Significance

From the comprehensive analysis of a large number of domestic and foreign studies, it can be seen that current research on slopes and retaining walls mainly focuses on soil pressure and seepage under certain assumptions. In this paper, based on the traditional research on soil pressure and the anti-slip and overturning stability of retaining walls, a new method for force and displacement analysis of slopes and retaining walls is proposed, which can obtain the solutions of stress distribution at each point inside the slopes and retaining walls under the corresponding boundary conditions. On this basis, Duncan Tensor and Hooke's principal structure models are adopted for the slope and retaining wall,

respectively, to obtain the strain and displacement solutions for the slope and retaining wall. The new theoretical solutions proposed in this paper can be used not only to study the damage forms of retaining wall overturning and foundation sliding processes but also to study the damage forms of retaining wall tensile bulging. It can provide a theoretical basis for the design of slope control engineering, and according to the different forms of stress, different forms, and materials of retaining walls can be adopted, and more economical, reasonable, and effective control methods can be derived.

## 3. Problem Formulation

### 3.1. Side Slope Problems

For a long time, the finite unit method has been applied in slope analysis. The calculation block diagram used in the finite element calculation is shown in Figure 1, and its boundary conditions are often of two types. The first type is displacement boundary conditions. The horizontal and bottom vertical displacements around the perimeter in Figure 1 are equal to zero; that is, when the displacement on the left (point *i*) is equal to zero, the displacement on the right (point *j*) must also be equal to zero. According to the definition of the displacement calculation:

$$u_i - u_j = \int_i^j \varepsilon_{i \to j} dl \tag{1}$$

where $u_i, u_j$ denotes the displacements at points *i* and *j* in the figure, $\varepsilon_{i \to j}$ denotes the strain in the *ij* segment, and dl denotes the integration along the *i* to *j* segments. From Equation (1), it can be seen that to make the *ij* segment displacement integral equal to zero, only the integration of strain in the *ij* segment is equal to zero; however, in the horizontal segment surface immediately below the bottom edge, the site should be dominated by compressive strain, and it is unlikely to produce tensile and compressive strain resulting in the sum of the *ij* segment displacement integral that is equal to zero.

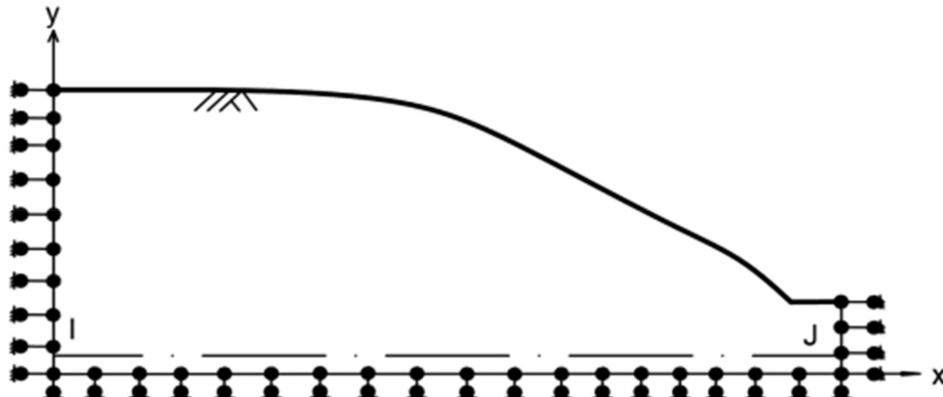

**Figure 1.** Displacement boundary condition chart of slope analysis.

The second type is the stress boundary condition. The boundary condition for the numerical analysis of the slope is shown in Figure 2. The far-field stress boundary condition is calculated according to elastic mechanics. At the peripheral stress boundary, the expression of the boundary condition is as follows.

$$\sigma_{xx}|_{x=0} = \sigma_{zz}|_{x=0} = \frac{v}{1-v} \gamma_y y; \; \sigma_{yy}|_{x=0} = \gamma_y y; \; \tau_{xy}|_{x=0} = \tau_{yz}|_{x=0} = \tau_{xz}|_{x=0} = 0 \tag{2}$$

where $v, \gamma_y$ are the Poisson ratio and specific gravity of the geological material and $\sigma_{xx}, \sigma_{yy}, \sigma_{zz}, \tau_{xy}, \tau_{yz}, \tau_{xz}$ are the positive and shear stresses, respectively.

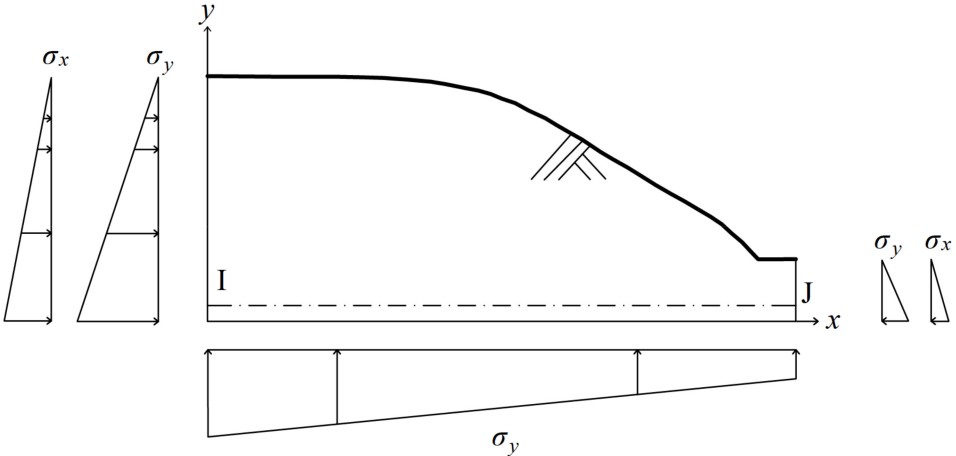

**Figure 2.** Stress boundary condition chart of slope analysis.

The physical meaning of Equation (2) is.

$$\sigma_{xx}|_{x=0} = \sigma_{zz}|_{x=0} = \sigma_3; \; \sigma_{yy}|_{x=0} = \sigma_1; \; \tau_{xy}|_{x=0} = \tau_{yz}|_{x=0} = \tau_{xz}|_{x=0} = 0 \tag{3}$$

where $\sigma_i, i = 1, 2, 3$ is the principal stress, and the corresponding $y$-value magnitude can be calculated by substituting Equation (3) into any current strength criterion. According to the inequality, when the numerical calculation results in the $y$-axis value greater than the $y$-value obtained according to the strength criterion, the damage has occurred at the site, i.e., with the increase in the calculated depth, the damaged area becomes increasingly large; however, the site becomes increasingly stable with increasing depth, which is not compatible with the site. A correct boundary condition corresponds to a correct solution; if the boundary condition is incorrect, the solution is incorrect.

### 3.2. Retaining Wall Problem

It can be seen from the traditional retaining wall design that the active earth pressure acting on the retaining wall is calculated with only the normal stress or horizontal force. From a mechanical point of view, even a vertical surface has both horizontal and vertical stress, so this stress calculation needs to be improved. In addition, the active earth pressure acting on the retaining wall is calculated with a damage angle of $(45^\circ + \varphi/2)$. The authors think this damage angle may only be suitable for old clay and gravel soils because not many soil slopes of $45^\circ$ are stable in the southern part of China.

To address the above problems, this paper applies the slip surface boundary method [19,20] and proposes a new numerical theoretical solution to obtain the stress and strain distribution of the slip body and retaining wall, based on which the stability evaluation description of the slip body and retaining wall is implemented.

## 4. Numerical Theoretical Solutions for Side Slopes and Retaining Walls
### 4.1. Basic Methods

For any material or object under the influence of boundary stress, when its shape is determined, its stress theory solution is well defined, and the result of the stress solution changes with a change in the boundary conditions.

Assuming that the medium satisfies the basic assumptions of elastodynamics and the stress solution satisfies the stress boundary, equilibrium, and coordination equations, the corresponding stress distribution solution is obtained. The method is based on elastic mechanics but can solve the problem of contact surface stress discontinuity. The method is applicable to the solution of the stress distribution in arbitrary geometry (including two-dimensional and three-dimensional problems); when the boundary conditions of the research object and the boundary stress distribution are not equal, the stress discontinuity solution can be obtained, and on this basis, the displacement discontinuity solution can

be obtained to solve the problem of stress-strain discontinuity in the damage process. The basic ideas and methods are as follows:

(1)    Establish descriptive equations for geometric features associated with the object of study based on precisely measured macroscopic geometric features;
(2)    When analyzing the distribution of the specific gravity of the object of study, establish the equation of its associated specific gravity distribution;
(3)    When analyzing the stress characteristics of the research object, the corresponding boundary condition stress equations are established according to the boundary conditions in different cases;
(4)    The representation of the stress equation is selected, and the corresponding constant coefficients are calculated, provided that the corresponding equilibrium equation, stress boundary condition equation, and coordination equation are satisfied for the object of study;
(5)    In the specific analysis of the force characteristics of the object of study, the damage characteristics are determined by combining the current strength code; the deformation characteristics of the object of study can also be compared with the relevant principal structure equations to obtain the behavior characteristics.

*4.2. Example of a Two-Dimensional Slope Retaining Wall*

Using the basic ideas presented above, the study of the theoretical solution to the two-dimensional slope plane strain retaining wall problem, as an example, is illustrated as follows:

For (i), based on the precisely measured macroscopic geometric features, the geometric characteristic descriptive equations associated with the object of study are established as in Figure 3, and they can be written in the form of $y = kx + b$ to represent the geometric characteristic descriptive equations for the boundaries of AB, BC, CD, DA, and FB, respectively. (Note: if the boundaries are in the form of curves, they can be described by the equations of curves).

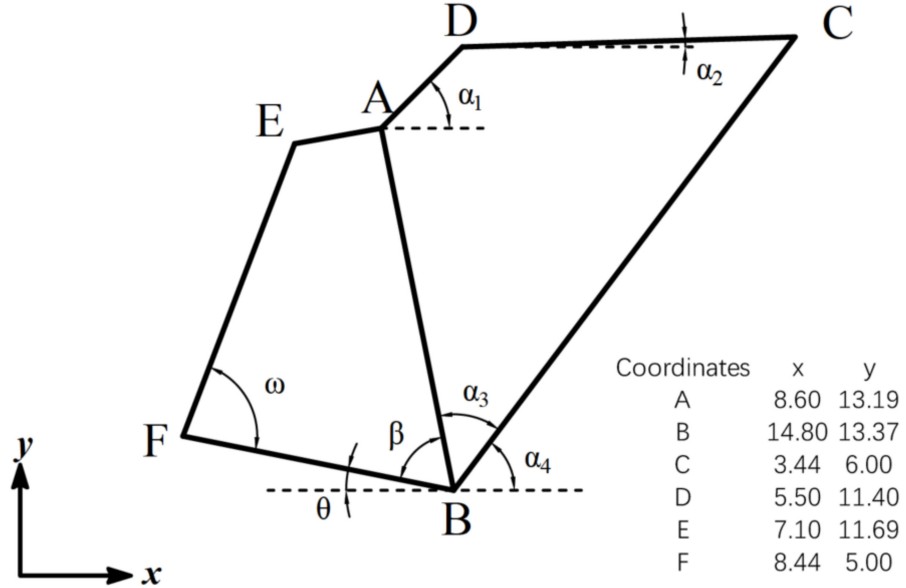

**Figure 3.** Calculation model of slope and retaining wall.

For (ii), when analyzing the weight distribution of the study object, its associated weight distribution equation is established as $\gamma_{w,x}, \gamma_{w,y}$.

For (iii), when analyzing the stress characteristics of the research object, the stress equations for the boundary conditions relative to the boundary conditions are established according to the boundary conditions in different cases; for the planar problems related to

Figure 3, such as horizontal stress ($p_x{}^{AB}$) and vertical stress ($p_y{}^{AB}$) on the AB boundary surface, the expressions are

$$p_x{}^{AB} = l\sigma_{xx}{}^{AB} + m\tau_{yy}{}^{AB} \tag{4}$$

$$p_y^{AB} = m\sigma_{yy}{}^{AB} + l\tau_{xy}{}^{AB} \tag{5}$$

where $l, m$ is the outer normal direction cosine on the *AB* boundary and $\sigma_{xx}{}^{AB}, \sigma_{yy}{}^{AB}, \tau_{xy}{}^{AB}$ is the boundary stress on the *AB* side. Under the condition of stress continuity, Equations (4) and (5) can describe all the different boundary condition stresses corresponding to Figure 3.

In the case of stress discontinuity, the stress at the discontinuity is not equal; however, the boundary condition stress forces need to be balanced along the horizontal and vertical directions.

For (iv), the representation of the stress equation is chosen, and the corresponding constant coefficients are calculated in the case that the object of study satisfies the corresponding equilibrium, stress boundary conditions, and coordination equations; in the two-dimensional condition, the stress expression is written (note: it can be changed according to different situations) as follows:

$$\sigma_{xx} = a_{1,1}x + a_{1,2}y + a_{1,3}x^2 + a_{1,4}xy + a_{1,5}y^2 + a_{1,6}x^3 + a_{1,7}x^2y + a_{1,8}xy^2 + \cdots \tag{6}$$

$$\sigma_{yy} = a_{2,1}x + a_{2,2}y + a_{2,3}x^2 + a_{2,4}xy + a_{2,5}y^2 + a_{2,6}x^3 + a_{2,7}x^2y + a_{2,8}xy^2 + \cdots \tag{7}$$

$$\tau_{xy} = a_{3,1}x + a_{3,2}y + a_{3,3}x^2 + a_{3,4}xy + a_{3,5}y^2 + a_{3,6}x^3 + a_{3,7}x^2y + a_{3,8}xy^2 + \cdots \tag{8}$$

The corresponding specific gravity equation (note: other representations are also possible) is written as follows:

$$\gamma_{w,x} = \gamma_{0,x} + a_{4,1}x + a_{4,2}y + a_{4,3}x^2 + a_{4,4}xy + a_{4,5}y^2 + a_{4,6}x^3 + a_{4,7}x^2y + a_{4,8}xy^2 + \cdots \tag{9}$$

$$\gamma_{w,y} = \gamma_{0,y} + a_{5,1}x + a_{5,2}y + a_{5,3}x^2 + a_{5,4}xy + a_{5,5}y^2 + a_{5,6}x^3 + a_{5,7}x^2y + a_{5,8}xy^2 + \cdots \tag{10}$$

where $a_{1,i}, a_{2,i}, a_{3,i}, a_{4,i}, a_{5,i}$ is the constant coefficient term, i is taken as zero and an integer, $\sigma_{xx}, \sigma_{yy}, \tau_{xy}$ is the stress and shear stress in the *x*- and *y*-axis directions, respectively, and $\gamma_{w,x}, \gamma_{w,y}$ is the specific gravity in the *x*- and *y*-axis directions, respectively.

Under gravity conditions, the general solution of the above stresses satisfying the equilibrium equation can be expressed as

$$\frac{\partial \sigma_{xx}}{\partial x} + \frac{\partial \tau_{xy}}{\partial y} = 0 \tag{11}$$

$$\frac{\partial \tau_{xy}}{\partial x} + \frac{\partial \sigma_{yy}}{\partial y} + \gamma_{w,y} = 0 \tag{12}$$

Then, the corresponding coefficients are zero, which is a necessary condition for the stress balance equation (note: Equations (11) and (12) can also be used to study the specific gravity relationship), and the following relationship can be obtained from Equation (11):

$$a_{1,1} + a_{3,2} = 0 \tag{13}$$

$$2a_{1,3} + a_{3,4} = 0 \tag{14}$$

$$a_{1,4} + 2a_{3,5} = 0 \tag{15}$$

$$3a_{1,6} + a_{3,7} = 0 \tag{16}$$

$$2a_{1,7} + 2a_{3,8} = 0 \tag{17}$$

$$a_{1,8} + 3a_{3,9} = 0 \tag{18}$$

. . .

From Equation (12), we have:

$$a_{3,1} + a_{2,2} = 0 \tag{19}$$

$$2a_{3,3} + a_{2,4} = 0 \tag{20}$$

$$a_{3,4} + 2a_{2,5} = 0 \tag{21}$$

$$3a_{3,6} + a_{2,7} = 0 \tag{22}$$

$$2a_{3,7} + 2a_{2,8} = 0 \tag{23}$$

$$a_{3,8} + 3a_{2,9} = 0 \tag{24}$$

. . .

By taking the specific gravity as a constant ($\gamma_{w,x} = 0, \gamma_{w,y} \neq 0 = \gamma$), the relative satisfaction of the boundary conditions, equilibrium equations, etc., can be solved for all constant coefficients, and when they are substituted into the stress expression, the stress solution of the object of study can be solved by Equations (6)–(8).

For (v), the damage characteristics of the object of study are determined by combining the current strength criterion with the specific analysis of its force characteristics; the corresponding intrinsic structure model is selected, the deformation characteristics are studied, and the behavior associated with them is further clarified by comparison with the field. The analysis process is as follows: first, the calculation process expressed in the paper is used to obtain the stress theory solution and then calculate the corresponding principal stress magnitude and substitute it into the strength criterion (e.g., the Moore–Coulomb criterion, Griffith criterion, etc.), to determine the damage state point, and then it is combined with the current strength theory to further determine the damage direction to determine the stress damage path. When the damage driving force is greater than its strength (i.e.,: stress discontinuity), there exists stress discontinuity and displacement discontinuity. While solving the stress discontinuity solution according to the above basic methods ((i) to (iv)), the damage path of the study object can be determined. The displacement continuity is solved by using coordinate rotation while considering the stress continuity, calculating the corresponding principal strain with the principal equation, and using the principal strain to obtain the strain at any point of the object of study. To solve the stress-strain problem under the premise of stress discontinuity, it is necessary to consider the deformation characteristics for calculation. By following the above solution steps, the stress-strain solution of the object of study during the whole damage process can be

solved, and the relevant physical–mechanical parameters in the theory can be corrected (i.e., inverse analysis) by comparing the actual situation in the field.

*4.3. Examples*

The stress theory solution of the research object is obtained by the above basic method and is used as an example for the retaining wall of the slope of a waste transfer station. The slope model is established, and ABCDEF in Figure 3 is the object of study. When analyzing the model, considering the interrelationship between the boundary conditions and the theoretical solution, the boundary conditions of the model in Figure 3 are described as  follows:

From the stress expression, which is a constant term, to $y^4$, there are 42 constant coefficients, which can be reduced to 22 constant coefficient expressions by the equilibrium Equations (11) and (12), and then based on the stress conditions on the given different boundaries, the corresponding constant coefficients can be determined.

The stress boundary conditions are satisfied at the *CD* and *AD* boundaries and are described as

$$p_x{}^{DC} = l^{DC}\sigma_{xx}{}^{DC} + m^{DC}\tau_{yy}{}^{DC} = 0 \tag{25}$$

$$p_y^{DC} = m^{DC}\sigma_{yy}{}^{DC} + l^{DC}\tau_{xy}{}^{DC} = 0 \tag{26}$$

Ten equations can be obtained using the DC boundary conditions, and the same AD side equation is consistent with the above Equations (25) and (26). The slope area of this study is composed of backfilled clay, and the angle between the BC side and the horizontal is $(45° + \varphi/2)$ ($\varphi$ denotes the friction angle within the soil) according to the traditional assumption that the surface has been damaged. The tangential stress along the surface is discontinuous, but the normal stress is continuous, and the tangential stress is in accordance with the strength discount method.  Then, the relationship between the tangential and normal stresses is as follows:

$$\tau_N^{BC} = \left(C + \sigma_N^{BC} tan\varphi\right)/f \tag{27}$$

where $\tau_N^{BC}, \sigma_N^{BC}$ are the tangential and normal stresses on the *BC* side and $C, \varphi, f$ are the cohesion, friction angle, and strength discount factor of the soil, respectively. The stress on the *AB* side is continuous; then, the equilibrium equation of the *ABCD* slip is seen below.

The forces are balanced in the horizontal direction as follows:

$$\int_{AB} p_x^{AB} dl + \int_{BF} p_x^{BF} dl = 0 \tag{28}$$

Vertical direction force balance:

$$\int_{AB} p_y^{AB} dl + \int_{BF} p_y^{BF} dl = W_{ABCD} \tag{29}$$

where $W_{ABEF}$ is the weight of the retaining wall unit width *ABCD*.

A computational model is established for the retaining wall, and *ABEF* in Figure 3 is taken as the object of study. For the analysis of the retaining wall model, the boundary conditions of the retaining wall *AB* and the continuity of the slip body boundary stress are considered, and the boundary conditions of the retaining wall model in Figure 3 are studied as follows.

Similarly, the stress expression, which is a constant term, is set to $y^4$, and the same process as that used to obtain *ABCD* is used to obtain the corresponding constant coefficients.

The stress boundary conditions are satisfied at the *EF* boundary, and the equations are of the same form as the *DC* and *AD* boundary Equations (25) and (26). However, the *AB*

boundary conditions are equal for the sliding body and the retaining wall stress boundary conditions; then, we have:

$$p_{x,h}^{AB} = p_{x,d}^{AB} \tag{30}$$

$$p_{y,h}^{AB} = p_{y,d}^{AB} \tag{31}$$

where $p_{x,h}$, $p_{x,d}$ and $p_{y,h}$, $p_{y,d}$ are the horizontal and vertical stresses of the sliding body and retaining wall at the *AB* boundary, respectively. The 20 equations can be obtained by using the *AB* and *EF* boundary conditions. In this study, the retaining wall is completely connected to the foundation, i.e., no damage occurs, and the stress is continuous. Then, the equilibrium equation of the *ABEF* slider is as follows.

The forces are balanced in the horizontal direction as follows:

$$\int_{AB} p_x^{AB} dl + \int_{BF} p_x^{BF} dl = 0 \tag{32}$$

Vertical direction force balance:

$$\int_{AB} p_y^{AB} dl + \int_{BF} p_y^{BF} dl = W_{ABEF} \tag{33}$$

where $W_{ABEF}$ is the weight of the retaining wall unit width *ABEF*.

*4.4. Computational Analysis*

4.4.1. Example of a Waste Transfer Station Project

The Shennong Creek Area Garbage Transfer Station Project is located in Fengjia Dagou, Group 1, Wulidui Village, Shennong Creek Area, Guandukou Town, Badong County, Hubei Province, and National Highway 209 crosses the west side of the site. The project covers an area of 1443.06 m$^2$, and the basic features of the garbage transfer station are as follows: the platform elevation is 283 m, the bottom elevation of the retaining wall is 274.8~283 m, and the slope height is 0~8.2 m (see Figures 4 and 5). The surface layer of the waste transfer station is red clay and was formed according to the I-I section (see Figure 5). The lower part of the transfer station is composed of T$_2$b$_2$ sandstone with high strength, uniaxial compressive strength 40~60 MPa, and rock inclination 260–300, and its retaining wall foundation is located above the weathered sandstone.

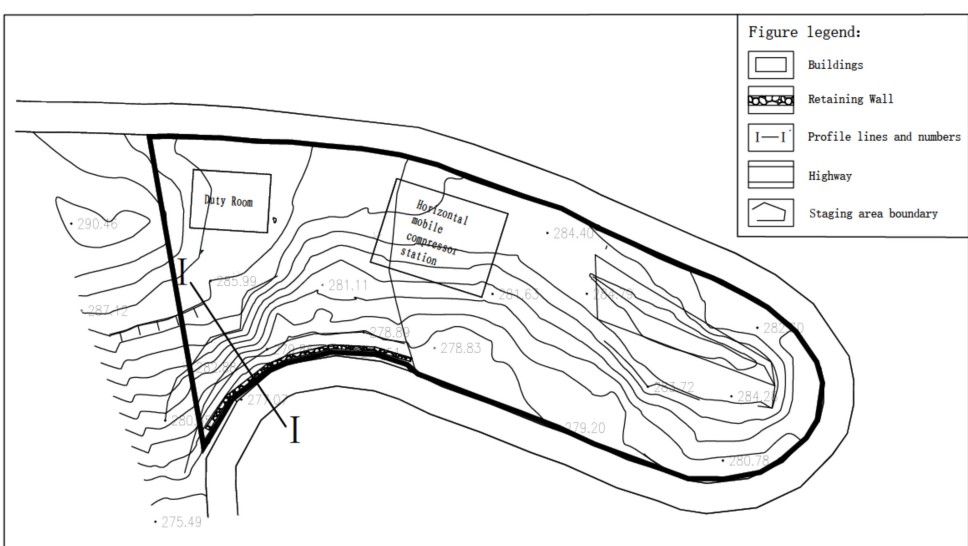

**Figure 4.** Plane plan of the transfer station area.

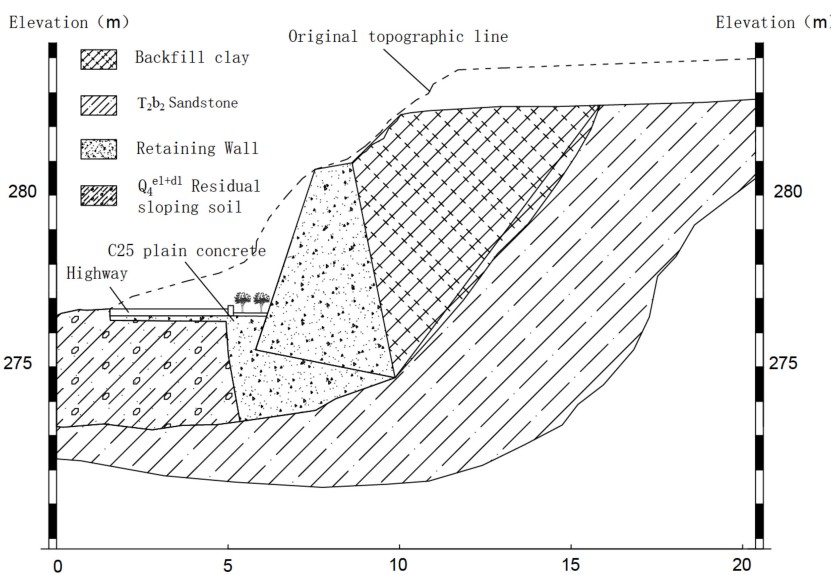

**Figure 5.** Section I-I of transfer station area.

### 4.4.2. Calculating the Model Dimensions

According to the I-I profile of the Guandukou waste transfer station, the specific gravity of the backfill clay of the waste transfer station is set to 19.6 kN/m$^3$, the friction angle is 18°, and the basic dimensions of the *ABCD* area of the slope are *AB* = 6.8 m, *BC* = 10.48 m, *CD* = 6.14 m, and *DA* = 2.12 m. The retaining wall *ABEF* is C25 plain concrete, and the basic dimensions are *AB* = 6.8 m, *BF* = 4.2 m, *EF* = 5.8 m, *AE* = 1.62 m, *EF* = 5.8 m, and *AE* = 1.62 m (see Figure 3). The specific gravity is set to 25kN/m$^3$.

### 4.4.3. Analysis of the Calculation Results
Stress Calculation Results of the Slope and Retaining Wall

According to the calculation of the slope retaining wall model, the coordinates corresponding to each point in the model are determined, the geometric boundary descriptive equations are established, the conditions associated with each boundary are determined, and the correlation coefficients under different discount factors ($f = 1.00, 1.50, 2.00$) are solved. The correlation coefficients are solved under different discount factors. The obtained coefficients are substituted into Equations (6)–(8) to obtain the distribution of $\sigma_{xx}, \sigma_{yy}, \tau_{xy}$ (see Figures 6–8) and the distribution of the principal stress $\sigma_1, \sigma_3$ (see Figures 9–11) (Note: Stress and cohesion are in kPa) at any point according to the corresponding coordinate points. Assuming that the peak stress of the retaining wall satisfies the Moore–Coulomb criterion, its friction angle is set to $\varphi = 40°$. According to the stress distribution characteristics of the retaining wall, the corresponding distribution of the cohesive force *C* values can be calculated, as shown in Figure 12. According to the magnitude of the *C* values, it can be determined where the retaining wall will be damaged first.

Slope Strain Calculation Results

Using the intrinsic structure relationship, the strain of the clay slope body is obtained according to the test, and it satisfies the Duncan–Zhang intrinsic structure model, whose basic equation is

$$\sigma_1 - \sigma_3 = \frac{\varepsilon_1}{a_1 + b_1\varepsilon_1} \Rightarrow \varepsilon_1 = \frac{a_1(\sigma_1 - \sigma_3)}{1 - b_1(\sigma_1 - \sigma_3)} \tag{34}$$

$$\sigma_1 - \sigma_3 = \frac{\varepsilon_3}{a_2 + b_2\varepsilon_3} \Rightarrow \varepsilon_3 = \frac{a_2(\sigma_1 - \sigma_3)}{1 - b_2(\sigma_1 - \sigma_3)} \tag{35}$$

where $\varepsilon_1, \varepsilon_3$ represents the first and third principal strains, respectively. According to the test results, $a_1$ is 0.0002, $a_2$ is 0.00012099, $b_1$ is −0.000056, and $b_2$ is 0.0002099. The

distribution characteristics of the principal strains of the slope under different reduction factors can be obtained (see Figures 13 and 14 and Section 4.4.3.3).

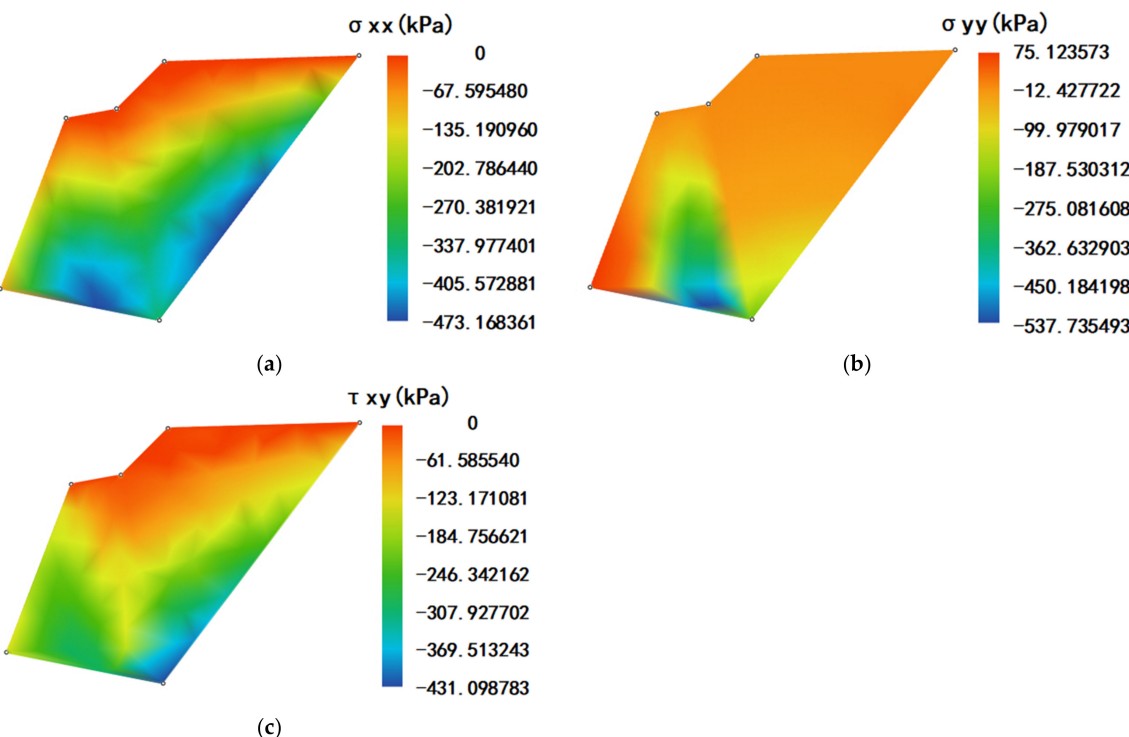

**Figure 6.** (**a**) Stress $\sigma_{xx}$ distribution of the sliding body and retaining wall when $f = 1.00$, (**b**) stress $\sigma_{yy}$ distribution of the sliding body and retaining wall when $f = 1.00$, (**c**) stress $\tau_{xy}$ distribution of the sliding body and retaining wall when $f = 1.00$.

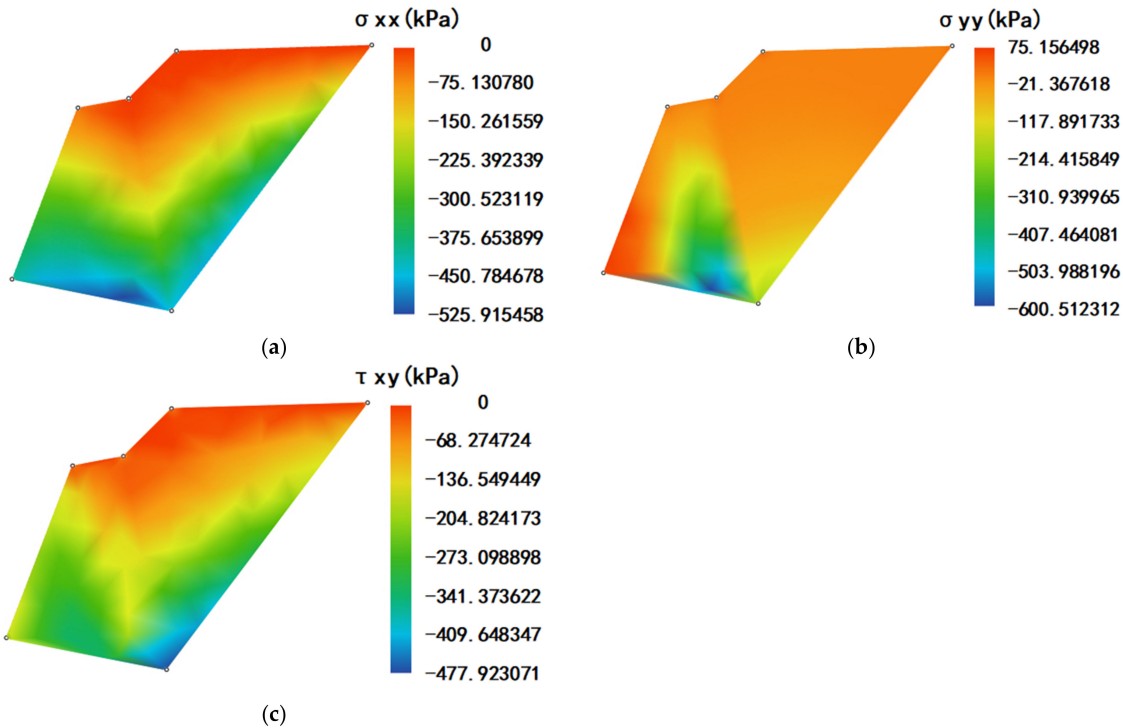

**Figure 7.** (**a**) Stress $\sigma_{xx}$ distribution of the sliding body and retaining wall when $f = 1.50$, (**b**) stress $\sigma_{yy}$ distribution of the sliding body and retaining wall when $f = 1.50$, (**c**) stress $\tau_{xy}$ distribution of the sliding body and retaining wall when $f = 1.50$.

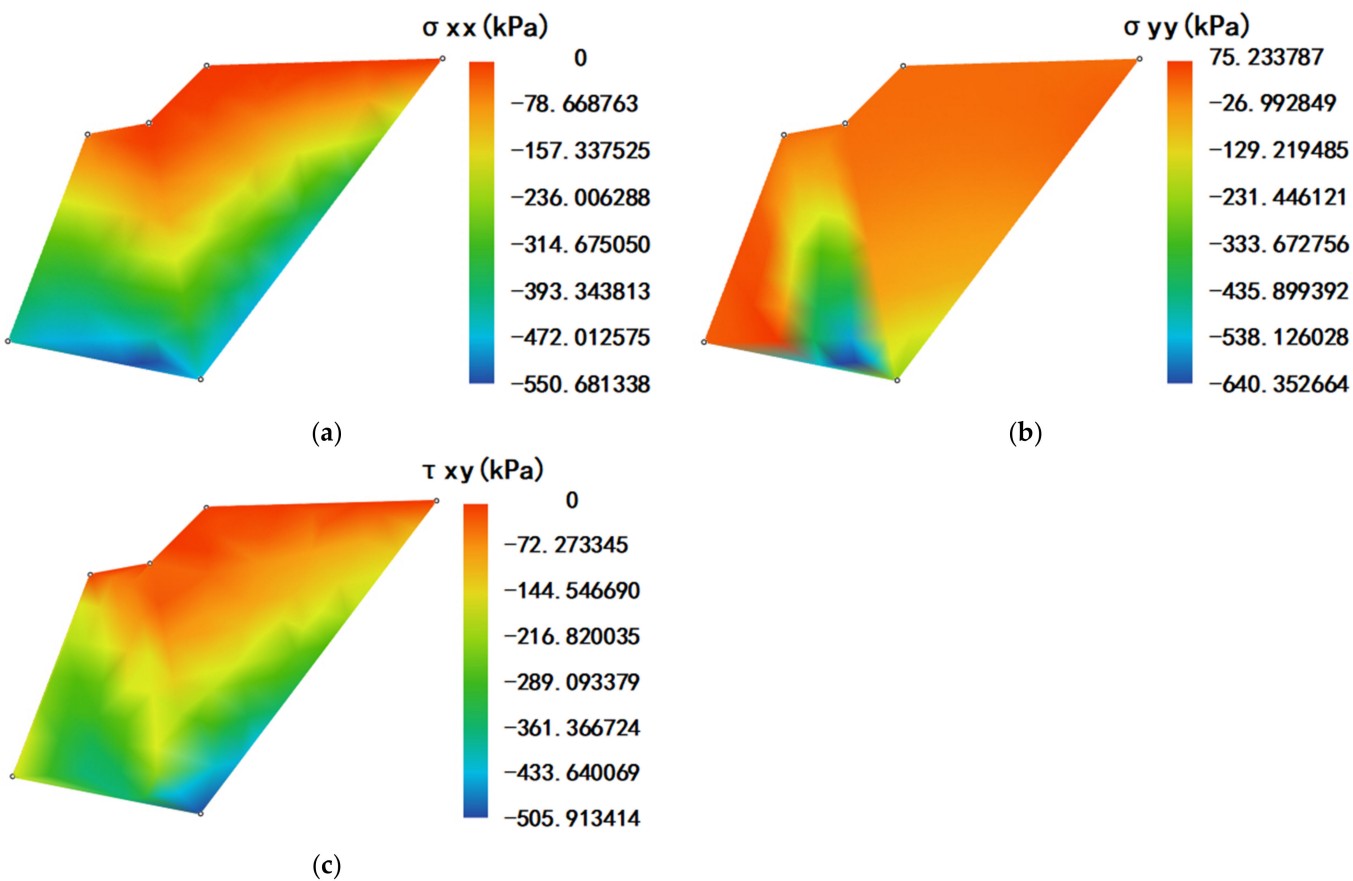

**Figure 8.** (**a**) Stress $\sigma_{xx}$ distribution of the sliding body and retaining wall when $f$ = 2.00, (**b**) stress $\sigma_{yy}$ distribution of the sliding body and retaining wall when $f$ = 2.00, (**c**) stress $\tau_{xy}$ distribution of the sliding body and retaining wall when $f$ = 2.00.

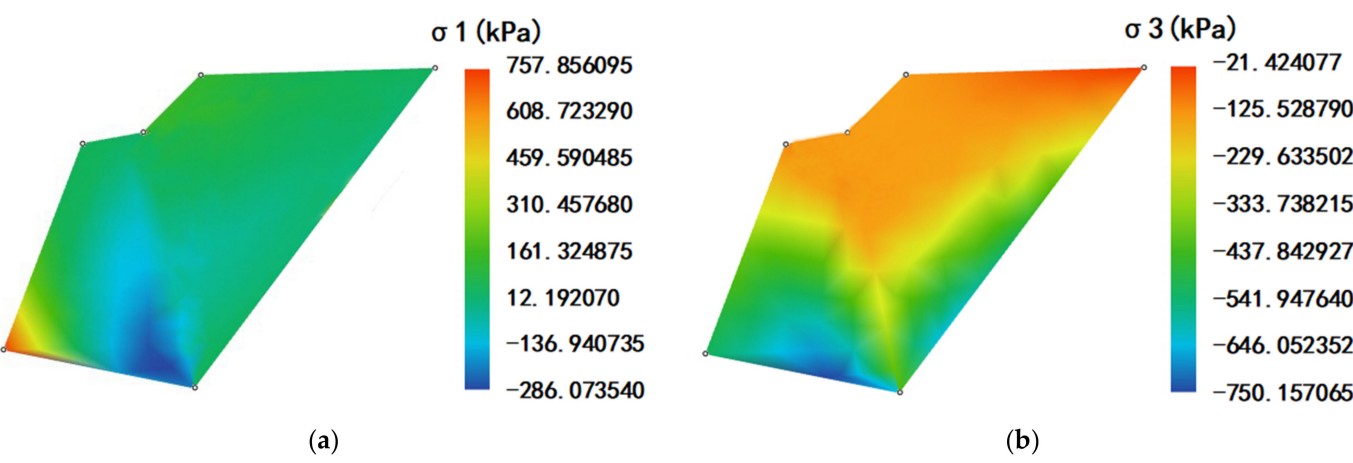

**Figure 9.** (**a**) Stress $\sigma_1$ distribution of the sliding body and retaining wall when $f$ = 1.00, (**b**) stress $\sigma_3$ distribution of the sliding body and retaining wall when $f$ = 1.00.

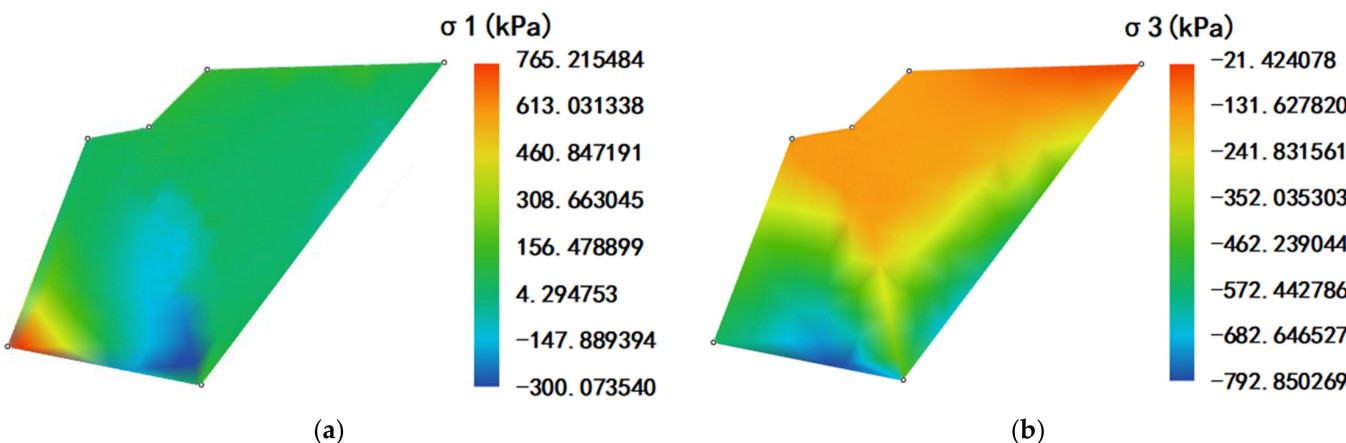

**Figure 10.** (**a**) Stress $\sigma_1$ distribution of the sliding body and retaining wall when $f$ = 1.50, (**b**) stress $\sigma_3$ distribution of the sliding body and retaining wall when $f$ = 1.50.

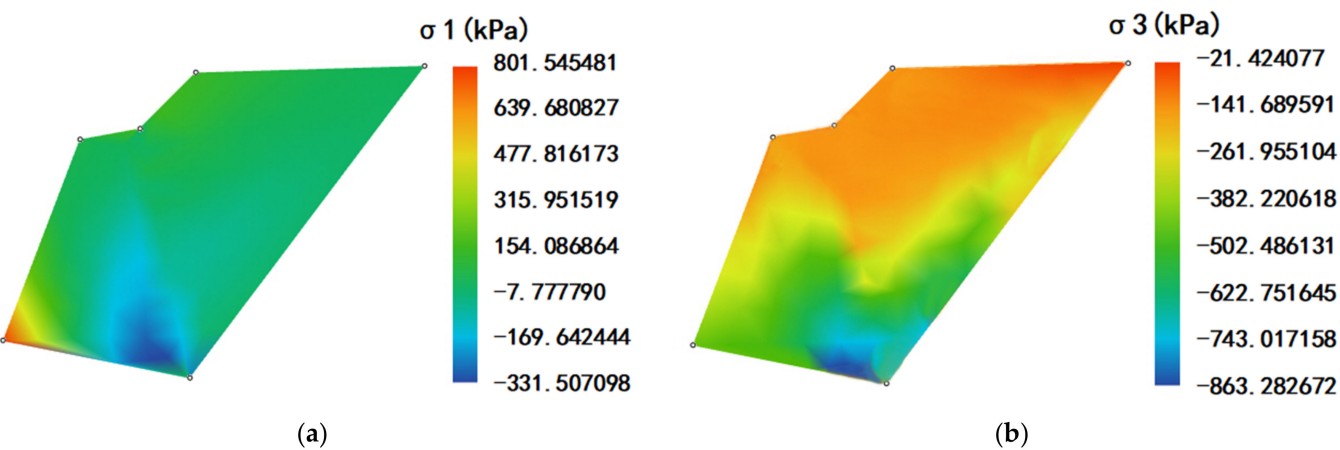

**Figure 11.** (**a**) Stress $\sigma_1$ distribution of the sliding body and retaining wall when $f$ = 2.00, (**b**) stress $\sigma_3$ distribution of the sliding body and retaining wall when $f$ = 2.00.

For a two-dimensional problem, the expression for the strain ($\varepsilon_{ij}$) in either direction of the rotation (rotation angle $\phi$) is

$$\varepsilon_{xx} = \varepsilon_1 cos^2\phi + \varepsilon_3 sin^2\phi \tag{36}$$

$$\varepsilon_{yy} = \varepsilon_1 sin^2\phi + \varepsilon_3 cos^2\phi \tag{37}$$

$$\gamma_{xy} = -\left(\varepsilon_{xx} - \varepsilon_{yy}\right) tan(2\phi) \tag{38}$$

where $\varepsilon_{xx}, \varepsilon_{yy},\ \gamma_{xy}$ indicates the strain and $\phi$ indicates the rotation angle.

The rotation angle $\phi$ is determined by the following equation:

$$tan2\phi = \frac{-2\tau_{xy}}{\sigma_x - \sigma_y} \Rightarrow \phi = \frac{1}{2}arctan\left(\frac{-2\tau_{xy}}{\sigma_x - \sigma_y}\right) \tag{39}$$

The calculated strain distribution obtained for each point of the sliding body is shown in Figures 16–18

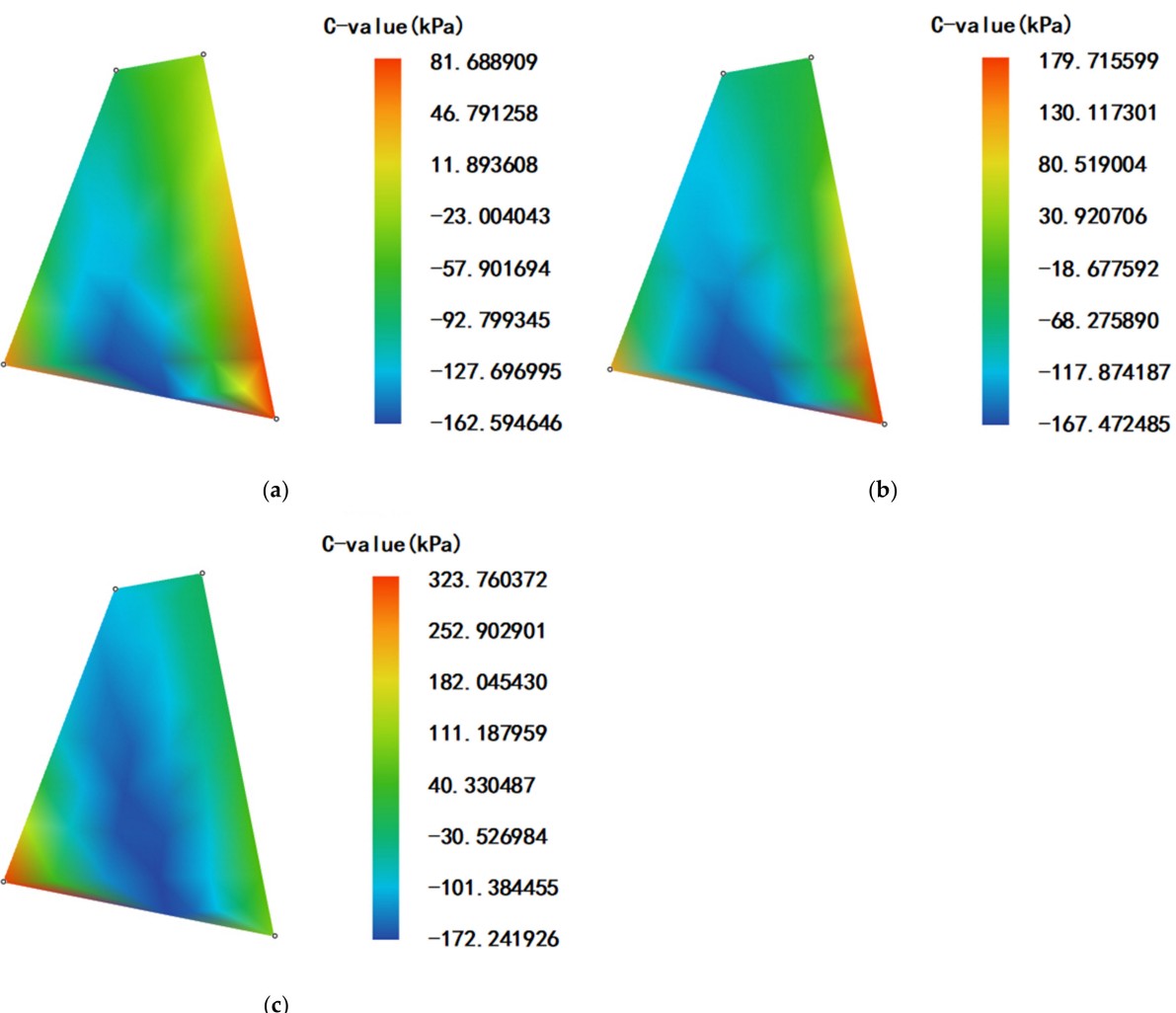

**Figure 12.** (**a**) Distribution diagram of the cohesive force C of the retaining wall when $f$ = 1.00, (**b**) distribution diagram of the cohesive force C of the retaining wall when $f$ = 1.50, (**c**) distribution diagram of the cohesive force C of the retaining wall when $f$ = 2.00.

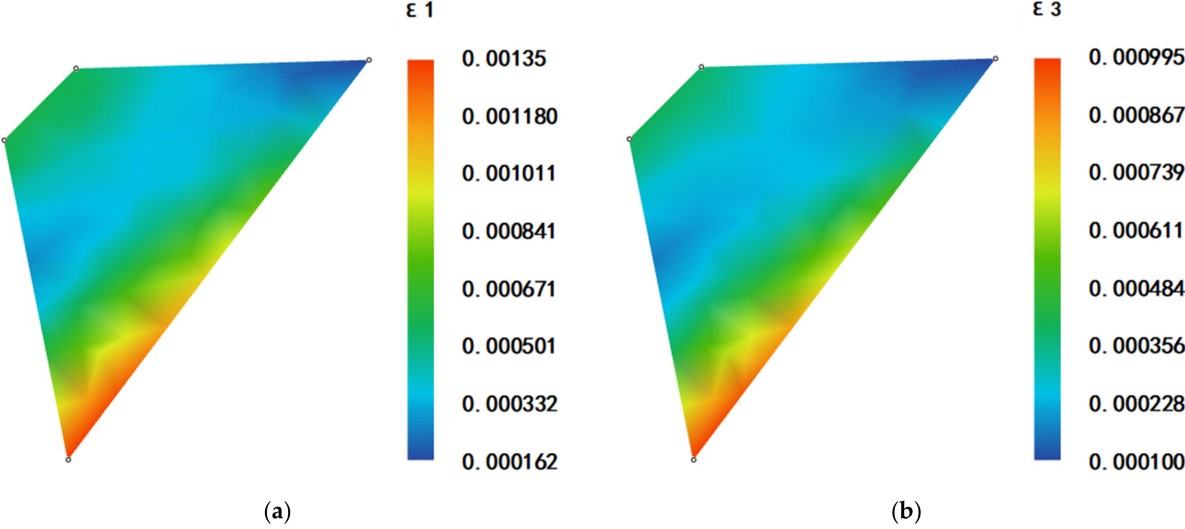

**Figure 13.** (**a**) Principal strain $\varepsilon_1$ distribution of the sliding body when $f$ = 1.00, (**b**) principal strain $\varepsilon_3$ distribution of the sliding body when $f$ = 1.00.

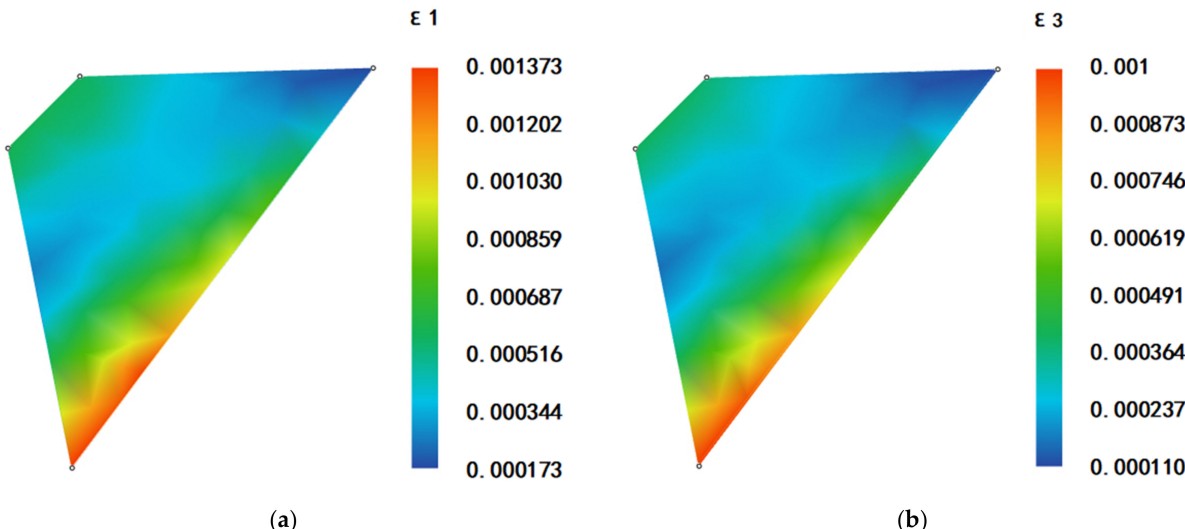

**Figure 14.** (**a**) Principal strain $\varepsilon_1$ distribution of the sliding body when $f$ = 1.50, (**b**) principal strain $\varepsilon_3$ distribution of the sliding body when $f$ = 1.50.

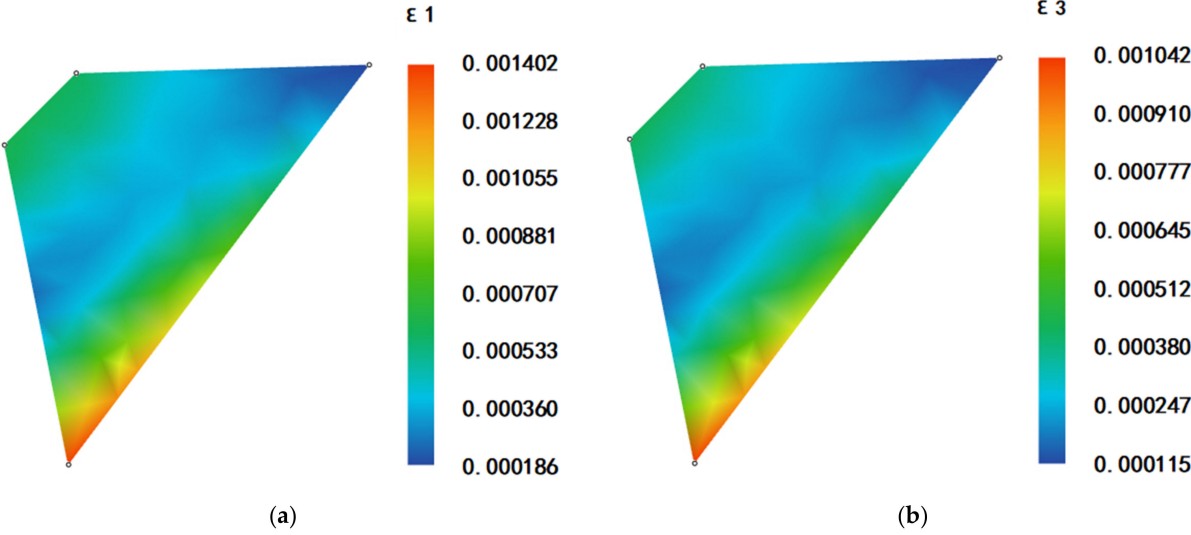

**Figure 15.** (**a**) Principal strain $\varepsilon_1$ distribution of the sliding body when $f$ = 2.00, (**b**) principal strain $\varepsilon_3$ distribution of the sliding body when $f$ = 2.00.

Results of the Strain Calculation for the Retaining Wall

Calculating the retaining wall strain can be considered a plane strain problem when $\varepsilon_z = 0$ and $\sigma_z \neq 0$. The general form according to Hooke's law is expressed as follows:

$$\varepsilon_{xx} = \frac{1}{E}\left[\sigma_{xx} - \mu(\sigma_{yy} + \sigma_{zz})\right] \tag{40}$$

$$\varepsilon_{zz} = \frac{1}{E}\left[\sigma_{zz} - \mu(\sigma_{xx} + \sigma_{yy})\right] \tag{41}$$

$$\gamma_{xy} = \frac{\tau_{xy}}{G}, \gamma_{yz} = \frac{\tau_{yz}}{G}, \gamma_{xz} = \frac{\tau_{xz}}{G} \tag{42}$$

where $G = \frac{E}{2(1+\mu)}$.

$E$ denotes the modulus of elasticity and is set to 300 MPa, $G$ expresses the shear modulus, and $\mu$ denotes the Poisson ratio, which is 0.11.

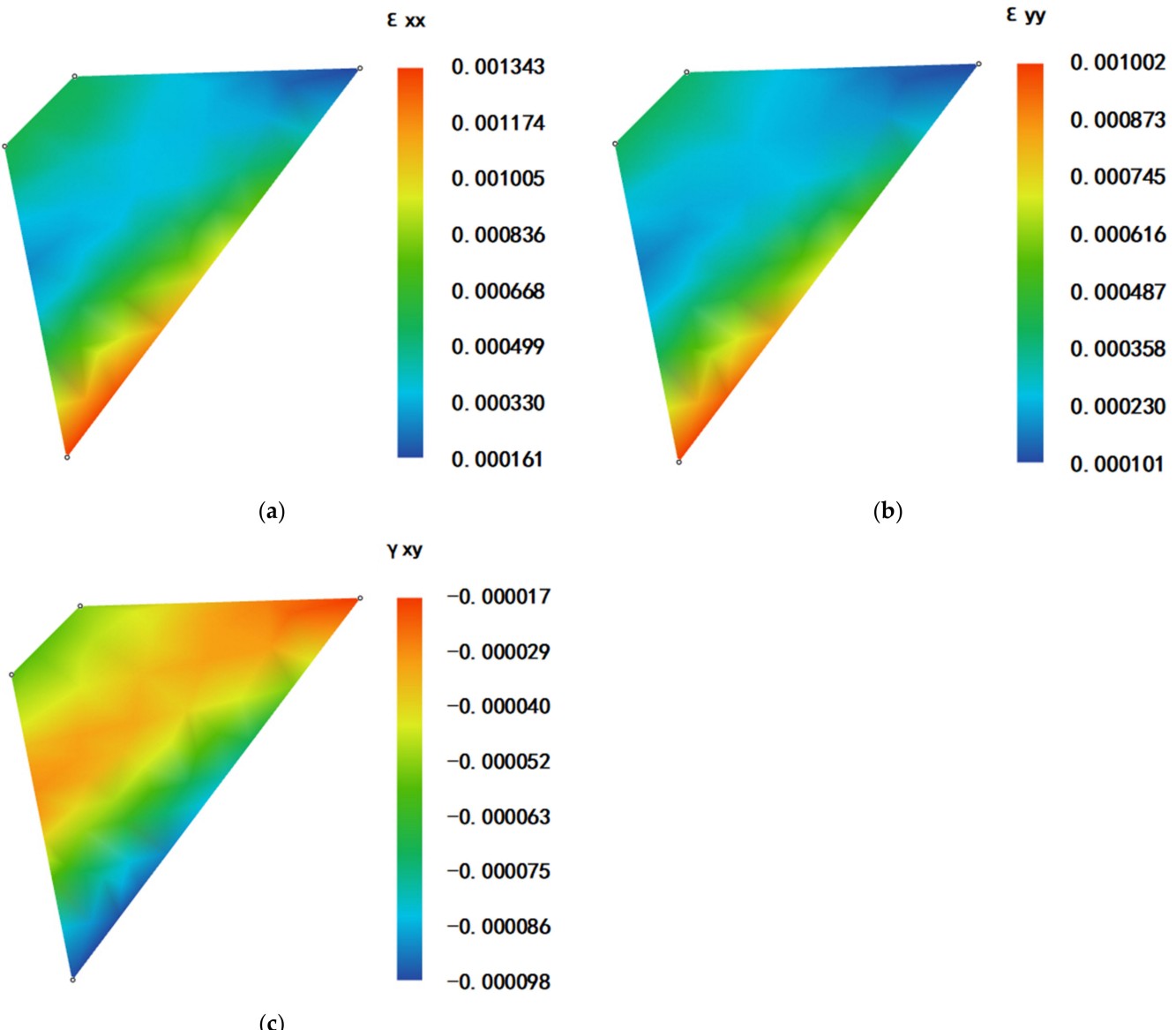

**Figure 16.** (**a**) Strain $\varepsilon_{xx}$ distribution of the sliding body when $f = 1.00$, (**b**) strain $\varepsilon_{yy}$ distribution of the sliding body when $f = 1.00$, (**c**) strain $\gamma_{xy}$ distribution of the sliding body when $f = 1.00$.

According to the above process, the strain distribution solutions of the retaining wall can be obtained, as shown in Figures 19–21. The corresponding principal strains are shown in Figures 22 and 23.

Retaining Wall Stability Analysis

From the calculation results of the retaining wall, it can be seen that the maximum tensile stress occurs at the corner of the wall and the slip body reduction factor becomes increasingly large ($f = 100,473.17$ kPa; $f = 150,500.82$ kPa; and $f = 200,530.71$ kPa), but its value falls within the strength range of C25 plain concrete; the absolute value of the compressive stress of the retaining wall increases with the slip body reduction factor and becomes increasingly larger ($-537.74$ kPa, $-600.51$ kPa, and $-640.35$ kPa), but it also falls within the strength range of plain concrete and foundation rock; for the C25 plain concrete retaining wall, the absolute value of the compressive stress increases with the slip body reduction factor. The absolute value of the compressive stress of the retaining wall increases with the discount factor of the sliding body ($-537.74$ kPa, $-600.51$ kPa, and $-640.35$ kPa), but it also falls within the strength range of plain concrete and foundation rock; for the C25

plain concrete retaining wall, the maximum value of the cohesion of the back-calculated points is 323.76 kPa under the condition that the friction angle is equal to 40 degrees, which also falls within the strength range of C25 plain concrete. The maximum value of cohesion is 323.76 kPa, which also falls within the cohesion value range of the C25 plain concrete strength, and the corresponding principal strain of the retaining wall is $10^{-3}$ level, which is within the peak strain range. Through the stress and deformation analysis of the retaining wall, the stress and strain at the point of the retaining wall are within the strength range, which means that the conditions under which damage occurs do not exist for the retaining wall. For the above problem, the finite unit method and Ansys software were used to calculate the error. The deviation in the stress-strain calculation for the backfill clay at the transfer station was less than 12%, and the deviation in the retaining wall result was less than 5%. Based on the analysis of the regulated retaining wall and the analysis results presented in this paper, the whole retaining wall is in a stable state, which is consistent with the results of many years of field operation.

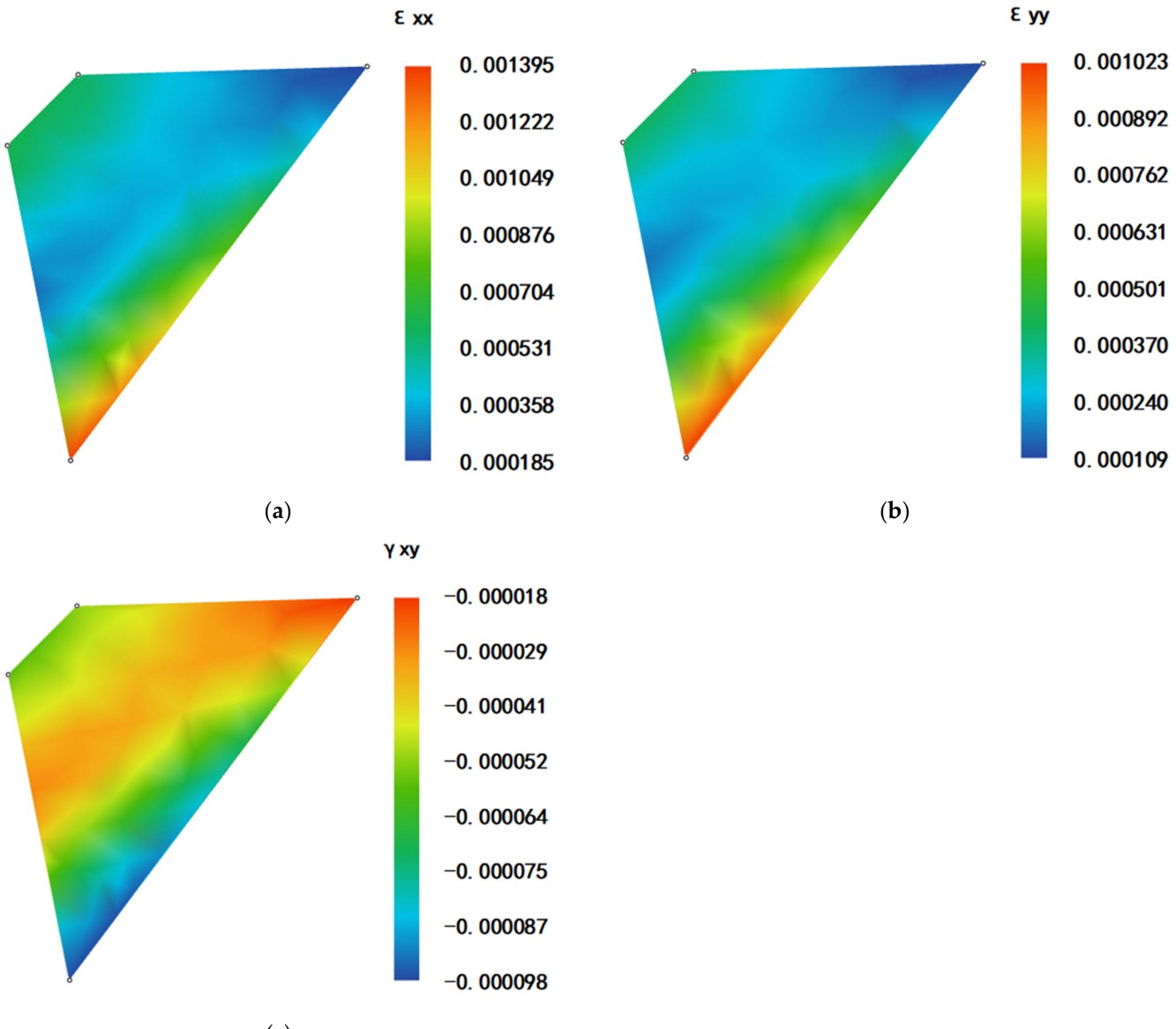

**Figure 17.** (**a**) Strain $\varepsilon_{xx}$ distribution of the sliding body when $f = 1.50$, (**b**) strain $\varepsilon_{yy}$ distribution of the sliding body when $f = 1.50$m, (**c**) strain $\gamma_{xy}$ distribution of the sliding body when $f = 1.50$.

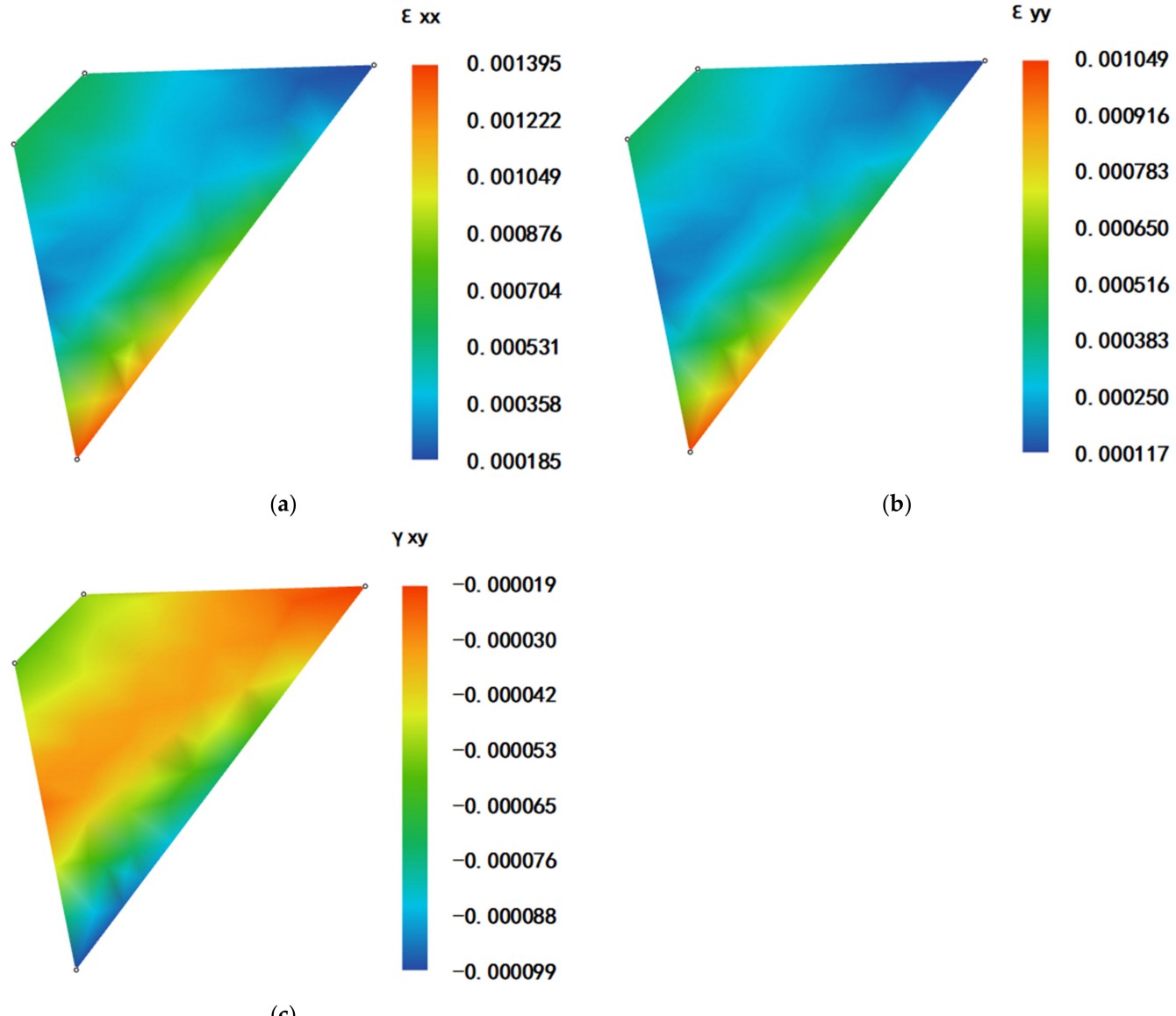

**Figure 18.** (**a**) Strain $\varepsilon_{xx}$ distribution of the sliding body when $f$ = 2.00, (**b**) strain $\varepsilon_{yy}$ distribution of the sliding body when $f$ = 2.00, (**c**) strain $\gamma_{xy}$ distribution of the sliding body when $f$ = 2.00.

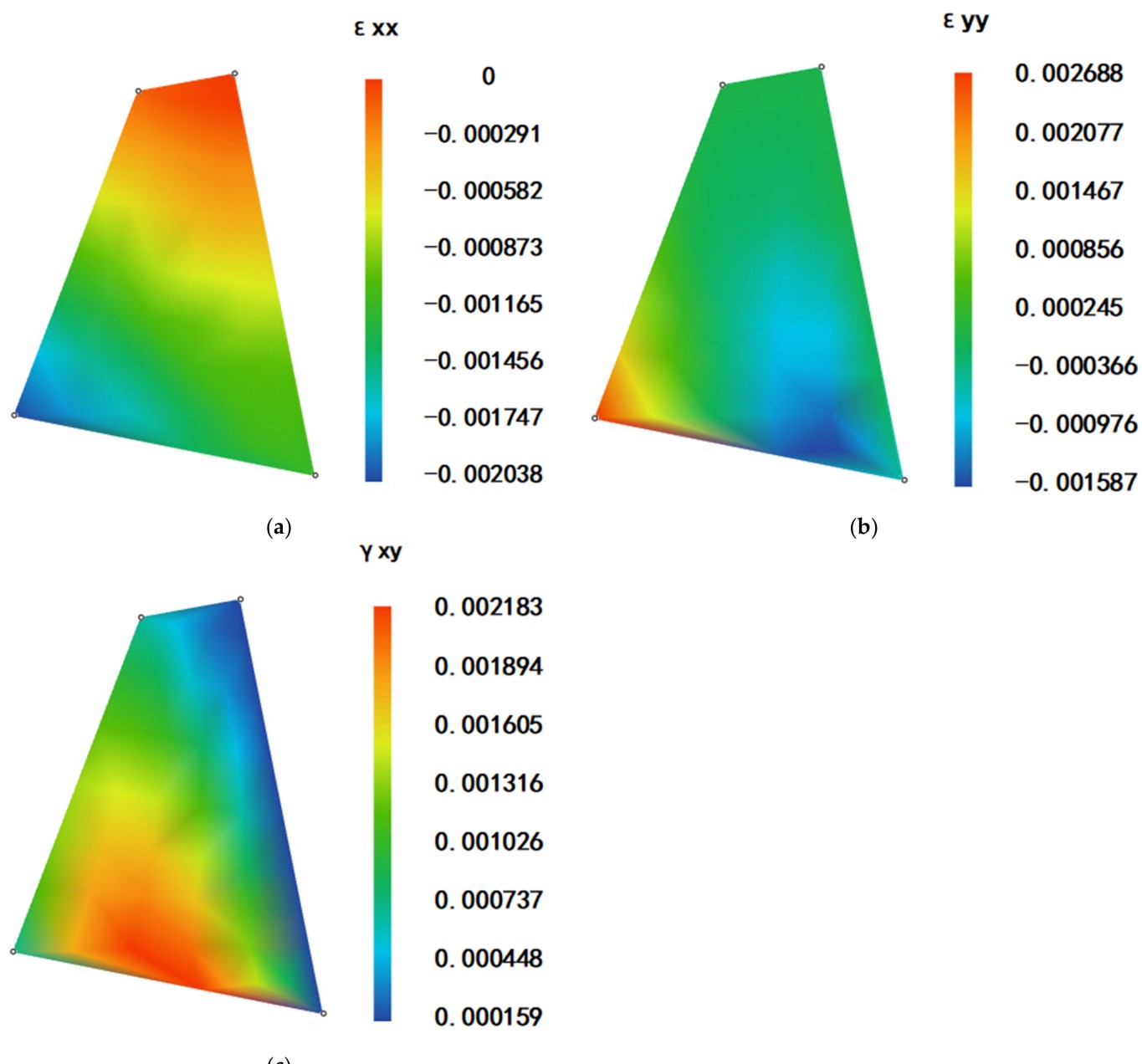

**Figure 19.** (**a**) Strain $\varepsilon_{xx}$ distribution of the retaining wall when $f$ = 1.00, (**b**) strain $\varepsilon_{yy}$ distribution of the retaining wall when $f$ = 1.00, (**c**) strain $\gamma_{xy}$ distribution of the retaining wall when $f$ = 1.00.

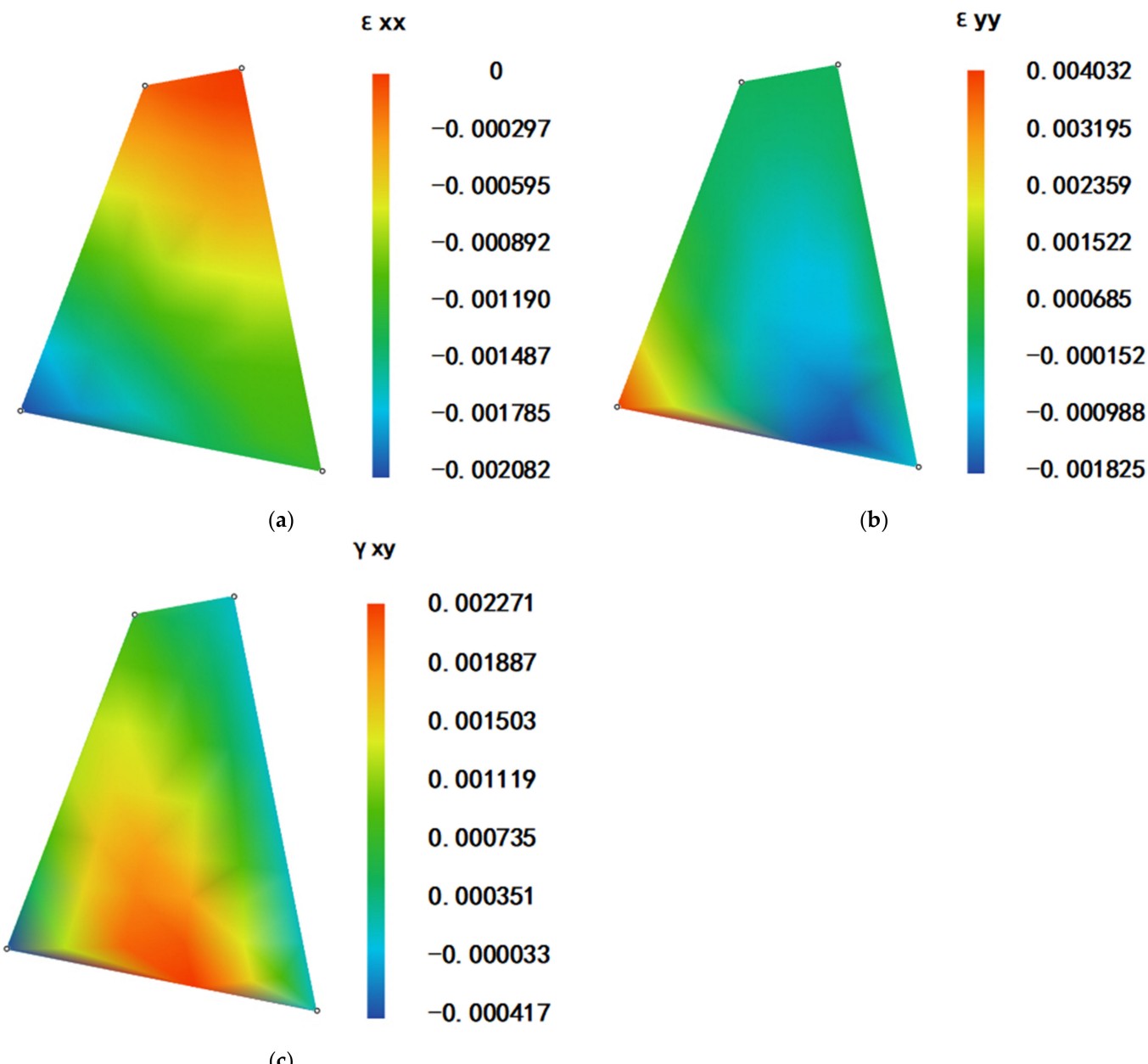

**Figure 20.** (**a**) Strain $\varepsilon_{xx}$ distribution of the retaining wall when $f$ = 1.50, (**b**) strain $\varepsilon_{yy}$ distribution of the retaining wall when $f$ = 1.50, (**c**) strain $\gamma_{xy}$ distribution of the retaining wall when $f$ = 1.50.

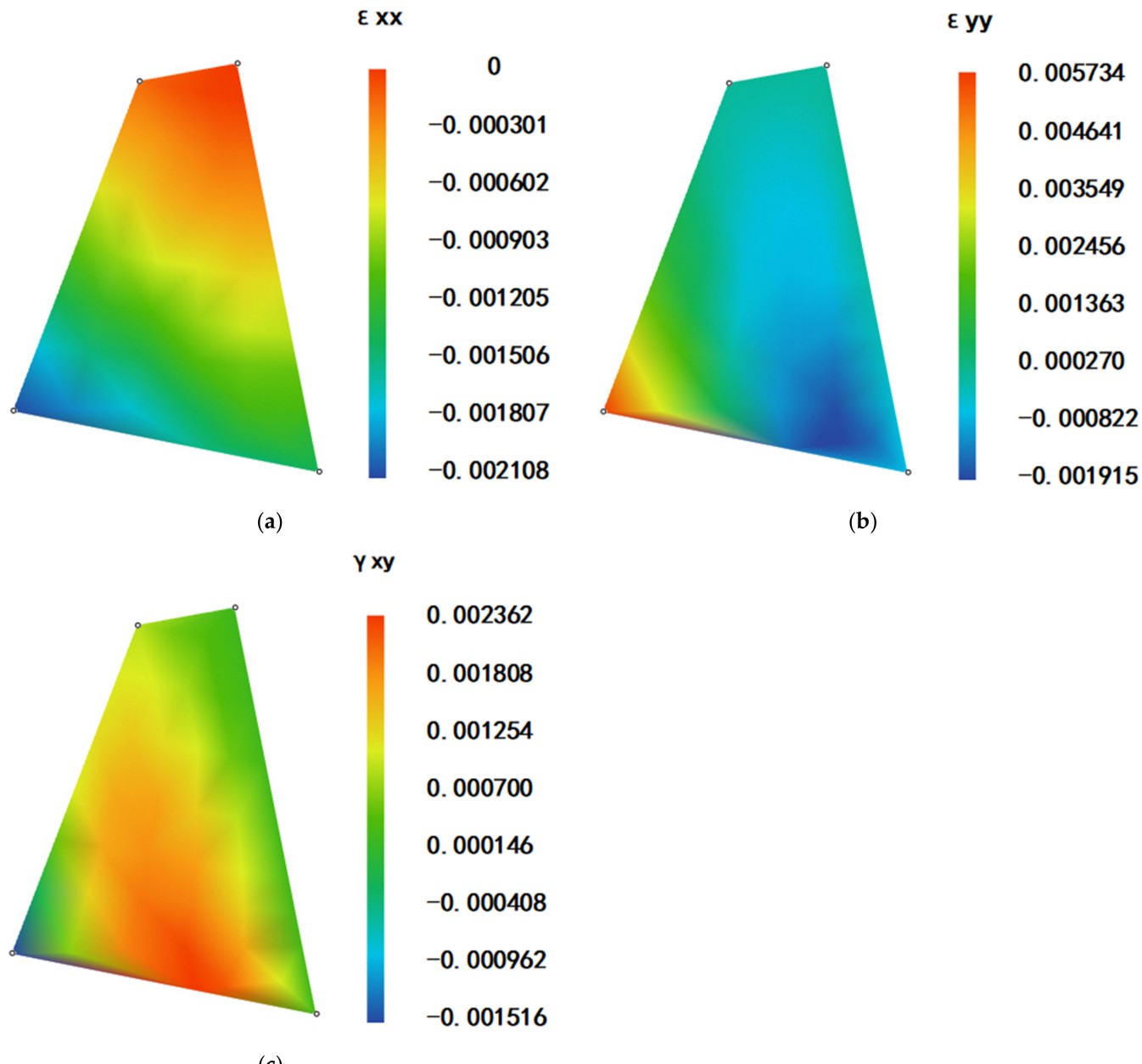

**Figure 21.** (**a**) Strain $\varepsilon_{xx}$ distribution of the retaining wall when $f = 2.00$, (**b**) strain $\varepsilon_{yy}$ distribution of the retaining wall when $f = 2.00$, (**c**) strain $\gamma_{xy}$ distribution of the retaining wall when $f = 2.00$.

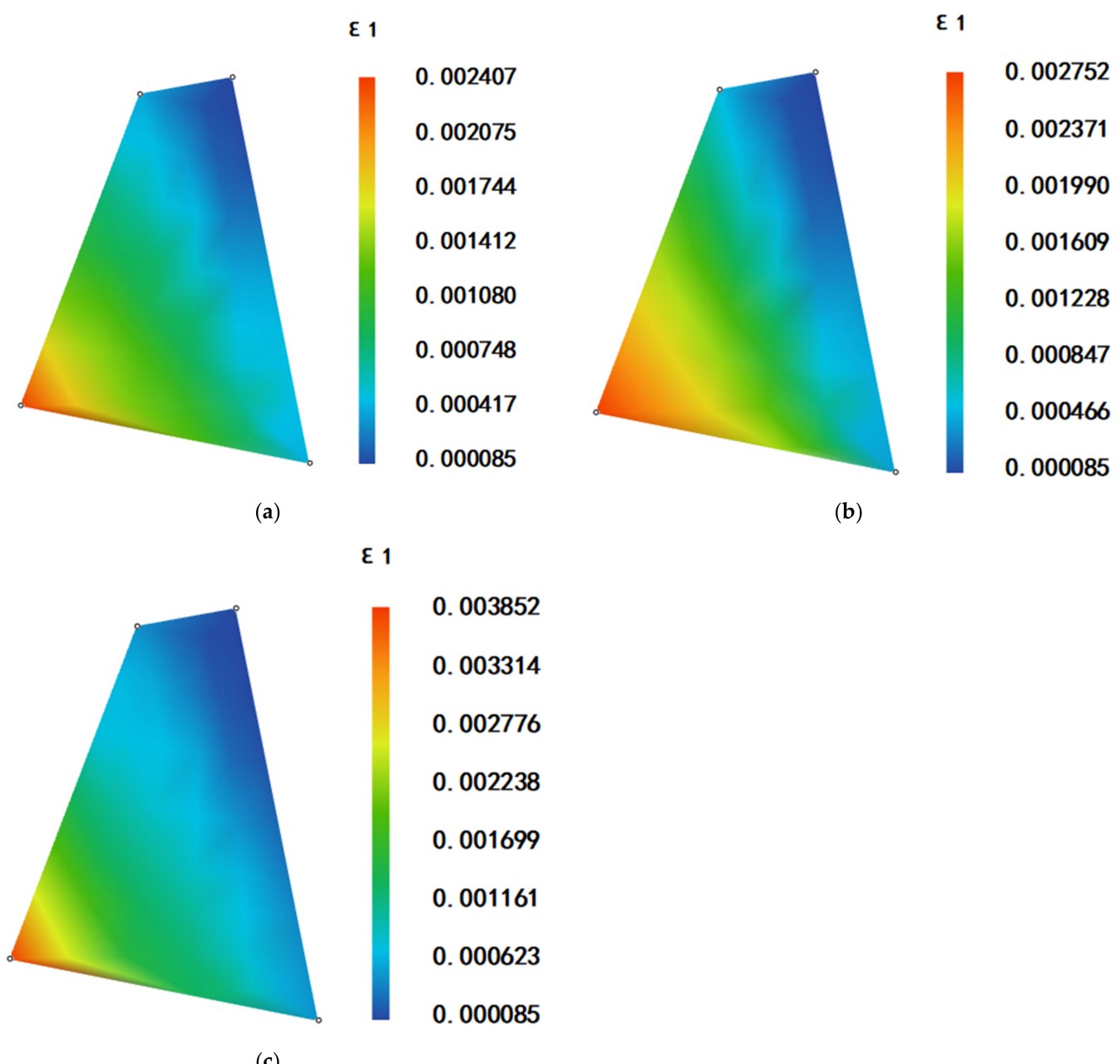

**Figure 22.** (**a**) Strain $\varepsilon_1$ distribution of the retaining wall when $f = 1.00$, (**b**) strain $\varepsilon_1$ distribution of the retaining wall when $f = 1.50$, (**c**) strain $\varepsilon_1$ distribution of the retaining wall when $f = 2.00$.

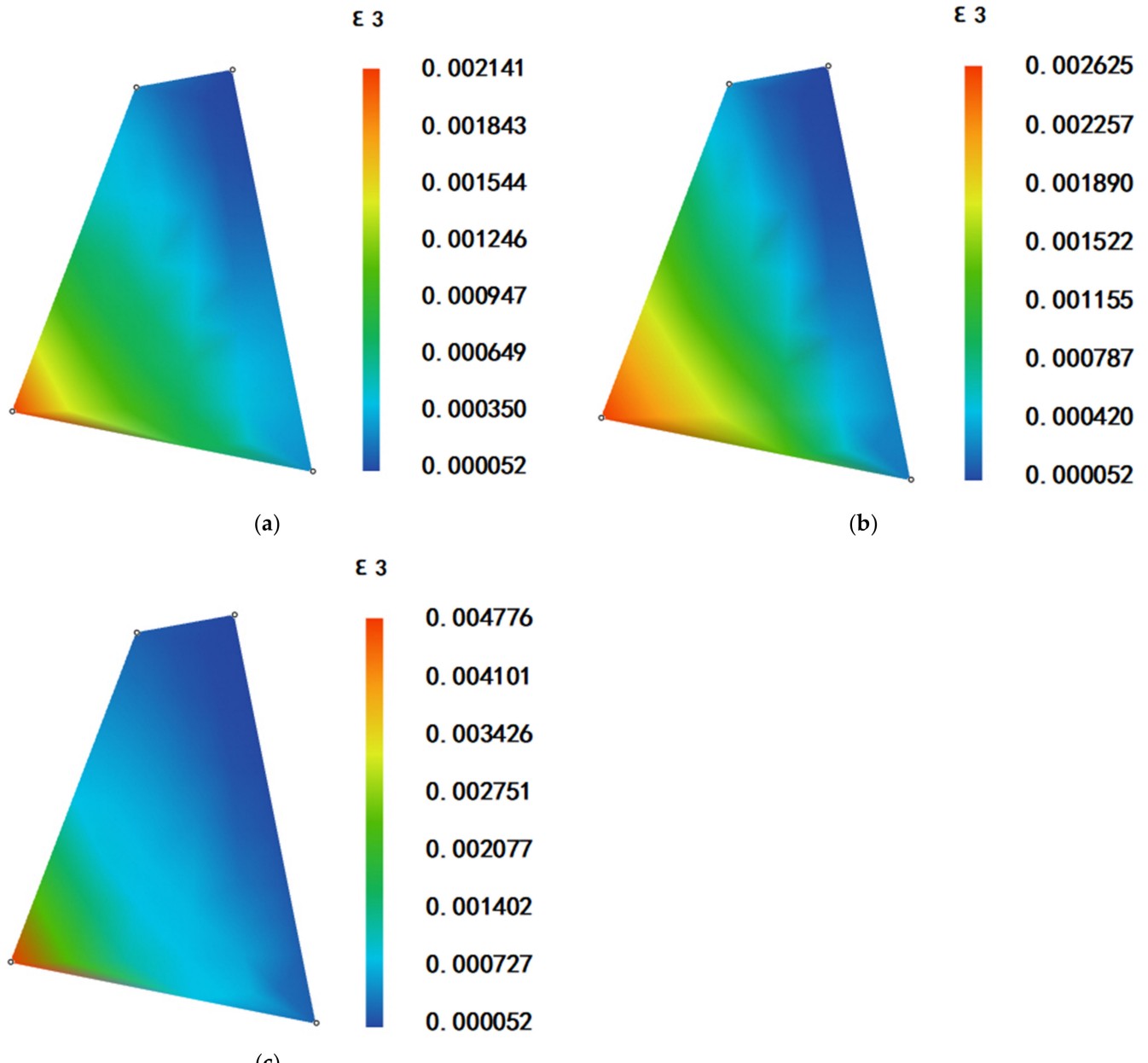

**Figure 23.** (**a**) Strain $\varepsilon_3$ distribution of the retaining wall when $f$ = 1.00, (**b**) strain $\varepsilon_3$ distribution of the retaining wall when $f$ = 1.50, (**c**) strain $\varepsilon_3$ distribution of the retaining wall when $f$ = 2.00.

## 5. Conclusions

Based on the literature [20] and the boundary conditions in this paper, the stress and strain distribution characteristics of backfill clay and retaining walls at a transfer station were obtained, and the following conclusions can be obtained from the solution characteristics:

(1) The backfill clay and retaining wall stresses at the transfer station are nonlinearly related to the coordinates. The calculation method proposed in this paper can provide a theoretical basis for the design of retaining wall anti-slip calculations and provide a design basis for slope stress monitoring. According to different retaining wall forms and materials, new retaining wall prevention and control methods can be derived.

(2) The numerical theoretical solution presented in this paper satisfies the basic assumptions of elastodynamics and is obtained under the assumption of continuous, isotropic, and continuous homogeneous stresses in the object of study. The results show that

the numerical theoretical solutions corresponding to different boundary conditions are also very different, i.e., the numerical theoretical solutions and the boundary conditions are closely related.

(3) Through the basic idea of this paper, it can be seen that the corresponding numerical theoretical solution can be obtained for any material (or object) given the boundary conditions, macroscopic characteristics, and specific gravity distribution of the object under study; the method can be applied to the study of stress distribution and damage processes of slopes, road foundations, tunnels, dams and other related materials under dynamic and static loading and unloading.

(4) In this paper, the shear stress problem with stress-strain discontinuity at the damaged surface is solved by applying the strength discount method, and it is noted that normal stress is continuous in this damaged surface region.

(5) This paper shows the practicality of the retaining wall design through the numerical theoretical solution results and provides a new point strength design method for a slip-resistant design.

(6) This paper shows the results of numerical theoretical solutions for retaining walls and slopes in two-dimensional planes. Considering the soil inhomogeneity in three-dimensional space, subsequent work will concentrate on how to apply it to three-dimensional space.

**Author Contributions:** Writing—review & editing: Y.L. Writing-original: W.S. and Y.L.; Data curation: W.S., H.Y., J.J. and L.L. All authors have read and agreed to the published version of the manuscript.

**Funding:** This work was supported by the National Natural Science Foundation of China (42071264, 41641027).

**Institutional Review Board Statement:** Not applicable.

**Informed Consent Statement:** Not applicable.

**Data Availability Statement:** The data used to support the findings of this study are available from the corresponding author upon reasonable request.

**Conflicts of Interest:** The authors declare that there are no conflict of interest regarding the publication of this paper.

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
