# Peer review of "A New Calculation Method of Force and Displacement of Retaining Wall and Slope"

_applsci, doi:10.3390/app13095806_

Round 1
Reviewer 1 Report
The purpose of this paper is to present a new calculation method for the determination of stress and strain in retaining walls. Despite its high scientific interest, the text is difficult and complex to read and understand. Part 3 is not easily understood. The authors should also go into more detail as it is the most important part. In addition, it is not clear to this reviewer what the novelty of this work is.
Line 128. "The far-field stress boundary condition is calculated according to the textbook of elastic mechanics." This sentence needs to be referenced.
Line 212. How many constant coefficients can be included in these equations?
All graphs must be scaled equally.
What are the units of the graphs?
Reviewer 2 Report
The manuscripts proposed a new calculation method for retaining wall and slope. The formulae are generally correct in line with slope stability. Some specific comments can be considered during the revise process.
(1) The format of the manuscript should be improved. In the current version, many Chinese symbols can be observed (e.g. Line 166, the digit 1 with a circle). The references are not cited in a professional manner. E.g. Chen Zhongda [1] can be replaced by Chen [1] – that is, only the family name is needed. This issue is applicable for all references citation in the draft.
(2) The proposed method should be validated by some existing methods to show its correctness. And also, the advantages of the proposed method should be discussed.
(3) The draft has too many figures. Some figures can be set as sub-figures. E.g. Figures 6 – 23.
(4) The basic assumptions of the proposed calculation method should be clearly stated. And the limitations of the method should also be pointed out.
(5) Although the authors pointed out the method is applicable for three-dimensional scenarios, a real slope or retaining wall is much complicated. For example, the soils are often non-uniform as pointed out by many references (e.g. paper entitled “Probabilistic stability analyses of undrained slopes by 3D random fields and finite element methods”). The non-uniformity of soils may affect the slope stability. This effect may be mentioned at proper place.
Reviewer 3 Report
The paper studies a new calculation method of force and displacement of retaining wall and slope. The study is interesting and useful to research community in current scenario. However, some minor corrections are needed to be incorporated as follow:
1. The quantitative results should reflect in the abstract. Abstract needs to be the overall summary of the study.
2. From line 96 to 108, the section can be separated under new heading Research and Significance.
3. The literature section needs to improved, Some recent works are suggested as below:
a) Hong, Y., Yao, M., & Wang, L. (2023). A multi-axial bounding surface p-y model with application in analyzing pile responses under multi-directional lateral cycling. Computers and Geotechnics, 157, 105301. doi: https://doi.org/10.1016/j.compgeo.2023.105301
b) Li, J., Chen, M., & Li, Z. (2022). Improved soil–structure interaction model considering time-lag effect. Computers and Geotechnics, 148, 104835. doi: https://doi.org/10.1016/j.compgeo.2022.104835
c) Zhang, C., Yin, Y., Yan, H., Zhu, S., Li, B., Hou, X.,... Yang, Y. (2022). Centrifuge modeling of multi-row stabilizing piles reinforced reservoir landslide with different row spacings. Landslides. doi: 10.1007/s10346-022-01994-5
d) Liu, C., Cui, J., Zhang, Z., Liu, H., Huang, X.,... Zhang, C. (2021). The role of TBM asymmetric tail-grouting on surface settlement in coarse-grained soils of urban area: Field tests and FEA modelling. Tunnelling and Underground Space Technology, 111, 103857. doi: https://doi.org/10.1016/j.tust.2021.103857
e) Wu, Z., Xu, J., Li, Y., & Wang, S. (2022). Disturbed State Concept–Based Model for the Uniaxial Strain-Softening Behavior of Fiber-Reinforced Soil. International Journal of Geomechanics, 22(7), 4022092. doi: 10.1061/(ASCE)GM.1943-5622.0002415
f) Wang, M., Yang, X., & Wang, W. (2022). Establishing a 3D aggregates database from X-ray CT scans of bulk concrete. Construction and Building Materials, 315, 125740. doi: https://doi.org/10.1016/j.conbuildmat.2021.125740
4Author needs to mention the novelty of work in the discussion section
The limitation of the work and future scope of the work must be added in conclusion section.
minor english editing in required
Round 2
Reviewer 1 Report
All the questions have been answered. No more comments are needed.